# Multiwall Rectangular Plates under Transverse Pressure—A Non-Linear Experimental and Numerical Study

**DOI:** 10.3390/ma16052041

**Published:** 2023-03-01

**Authors:** Gilad Hakim, Haim Abramovich

**Affiliations:** Faculty of Aerospace Engineering, Technion—Israel Institute of Technology (I.I.T.), Haifa 32000, Israel

**Keywords:** rectangular plate large deflection, Föppl–von Kármán equations, multiwall plate, nonlinear load–deflection curve, vacuum chamber loading test

## Abstract

Large deflection of rectangular plates under transverse pressure is described by Föppl–von Kármán equations, which have only approximated solutions. One of these methods is the separation into a small deflection plate and a thin membrane described by a simple third order polynomial expression. The present study presents an analysis to obtain analytical expressions for its coefficients by using the plate’s elastic properties and dimensions. To validate the non-linear relationship between the pressure and the lateral displacement of the multiwall plate, a vacuum chamber loading test is used to measure the plate’s response, with a large number of plates and length–width combinations. In addition, to further validate the analytical expressions, several finite element analyses (FEA) were performed. It has been found that the polynomial expression fairly describes the measured and calculated deflections. This method allows the prediction of plate deflections under pressure as soon as the elastic properties and the dimensions are known.

## 1. Introduction

The problem of large deflection of plates has attracted much attention since the end of the 19th century, due to its technical importance. Unfortunately, this problem has been found to be difficult to solve. The difficulties were raised at an early stage with Föppl equations for large deflections of membranes [1], where no closed-form solution was found for the rectangular membrane case. In 1910, this equation set was enhanced by Theodor von Kármán [2] to include the bending resistance of plates. 

The Föppl–von Kármán equation set has challenged many researchers over the years. Nevertheless, only approximated solutions were developed, most of which are rather difficult to implement.

It was August Föppl himself who suggested an approximate approach. This approach is mentioned in Timoshenko [3] (p. 423) footnote 1, which mentions Föppl’s “Drang und Zwang” [4] (p. 345). The approach is that the transverse distributed load *q* on the plate can be separated into two parts: *q* = *q*_1_ + *q*_2_. The first part *q*_1_ is balanced by the plate’s bending and shearing resistance, which are calculated through the plate small deflection linear theory. The second part *q*_2_ is balanced by the large deflections in-plane membrane forces only. Using the mid-point deflection *w*, this approximation is written as:(1)$$q=q_{1}+q_{2}=A\cdot w+B\cdot w^{3}$$

The plate’s small deflection coefficient *A* has been calculated in many previous studies, with most of them using a summation of the Fourier series. The large deflection coefficient *B*, however, has no exact solution, as it is ascribed to the difficult Föppl’s membrane problem [1].

This expression (1) for the load–deflection behavior is quoted by many sources, such as Timoshenko [3] (p. 424) for square isotropic plates, Ugural [5] (p. 358), Wang and El-Sheikh [6] (p. 816), as well as many others.

One of them is Riber [7], who had suggested a “combined analytical solution” (see p. 71 in [7]) to find the constants in Equation (1). He used an energy method to obtain rather complex expressions for the coefficients *A* and *B* (see Equation (1)). He also presented simplified expressions for the constant *B* but with some internal inconsistencies. Riber [7] assumed in-plane immovable edges, which is not the case to be presented in the present study—see the BCs (Boundary Conditions) discussion later in the paper. Nevertheless, his *B* equation has inspired the presentation of a better expression for the coefficient *B* later in the present study.

Awrejcewicz et al. [8] present many efficient numerical methods that can be used to calculate specific rectangular plates without orthotropic and transverse shear behaviors.

Battaglia et al. [9] analyze orthotropic membranes and show that the load is proportional to $w^{3}$, where the proportion coefficient can be compared to our cases.

Maier-Schneider et al. [10] found the membrane proportion coefficient with both the improved analytic energy method and the finite element method with a good agreement.

Niyogi [11] solves the simply supported orthotropic plate problem with an approximate Galerkin–Bubnov procedure. The resulting *q* = *Aw* + *Bw*³ expression is verified for the isotropic plate only. The edge in-plane BCs are immovable.

Wang et al. [12] also solve the static behavior of flat glass plate and present load–deflection data graphically. No explicit mathematical expression is given for that, although it can be extracted from the graph. Here, the in-plane BCs are totally immovable.

References [13,14,15] provide further theoretical elasticity basis on the present investigated topic.

During the literature survey, several sources were found referring to tests and calculation methods of plates’ large deflection. In order to compare the results of these papers, it was necessary to normalize the various data to a common comparable structure. The structure was a thin square isotropic plate with movable edges and an evenly distributed transverse load. The coefficients of Equation (1), *A* and *B*, were calculated considering the plate dimensions and the material properties. The result of this comparison has shown the considerable variability of the coefficient *B*. This variability was unexpected since most of the data was based on real laboratory tests that should respond in a similar way. This may demonstrate the fact that it is not easy to correctly measure this property. A full description of the comparison with a suggested explanation is presented in Appendix B.

## 2. Multiwall Plates

The structure of multiwall plates consists of two thin face sheets separated by an internal structure of ribs and walls. The plate is usually produced by extrusion, in which a melted material is pressed through a die with the required shape. The materials used are aluminum and various plastics. The result is a thick, endless plate with a fixed cross-section shape along the extrusion direction and width according to the equipment size. The plate is then cut to the desired length.

The plate is made of polycarbonate (PC), which is a tough transparent plastic. A typical 16mm PC plate can be seen in Figure 1.

The main application of PC multiwall plates is the glazing of architectural spaces, where both natural light and weather protection are required. As a result, the plates are exposed to wind and snow loads, which they must safely resist.

Currently, no publication describes the general performance of these plates, except manufacturer’s datasheets, which are very limited to specific products and specific applications. The available publications about latticed structures relate to specific shapes, such as triangles and trapezoids, and not a general approach as presented here, which can be considered novel.

The available approximated solutions for large deflections of plates are generally rather complicated, and in many cases, they involve a computational process, which is not straight-forward for field engineers. The non-linear nature of load–deflection curves is not easily represented in these solutions. Most of the research works already done do not cover the full complexity of the multiwall plates, which are shear, deformable, and orthotropic. Therefore, an engineer who needs to design a system with multiwall plates will probably face serious difficulties. This situation justifies this paper, which yields a first rough guess of these plates’ performance.

To calculate the multiwall plate response to distributed load, it is necessary to know the plate’s equivalent elastic properties, its dimensions (length–width), and the boundary conditions. Looking at the multiwall structure, it is obvious that the plate is orthotropic for both bending and tension, and its cross-section is transverse shear deformable. In the present study, it is assumed that all necessary equivalent elastic properties are already known. One should note that a procedure to obtain these equivalent properties of the plate is presented in Hakim and Abramovich [16].

## 3. Axes System

The axes directions are defined as shown in Figure 2, where axis *x* is the extrusion direction and *z* axis is normal to the plate surface:

The origin location of the axes may be set to any convenient place.

## 4. Boundary Conditions (BCs)

For small deflections analysis, the assumption is that all in-plane stress, strains, and deflections are negligible. Therefore, the BCs, here, ignore the in-plane conditions. The most commonly used BCs are: Free (F)-no restrictions, Simply Supported (S)-no *z* deflection but free rotation (no bending moments), and Clamped (C)-no *z* deflection and no rotations (zero slope). S and C are the two extremes of the more complicated BC-flexible rotation support, which is rarely used.

For large deflection analysis, the in-plane BCs must be considered. The two most common BCs are: Immovable (I)-the plate edge is fixed to the support and Movable (M)-the plate edges are allowed to move. It is necessary to specify both in-plane movement directions: normal to the edge and parallel to the edge. The BC used later here is SSSS-M, in which the four S stands for the four plate sides simply supported, and the M stands for the movable edges in both normal and parallel directions.

The movable (M) condition requires additional attention. When the Föppl’s approximation is used, the plate in a large deflection regime is a membrane. Its deflection on movable boundary conditions should then be calculated. However, a well-known property of a membrane is that it cannot sustain in-plane compression forces, as it immediately wrinkles. However, a real plate does resist compression, as it has a bending rigidity. We, therefore, have to analyze a membrane with movable edges, which may have compression stress. Mathematically, it is possible (with the known difficulties of Föppl equations), but other practical problems would appear. Since this transversely loaded M membrane is not a common case in the literature and, perhaps, even physically not possible, no previous scientific papers that would suggest a possible solution were found. Nevertheless, an expression is suggested later in this paper.

## 5. Methods

### 5.1. A-B Analytical Prediction

The expressions that would predict the values for the coefficients A and B are displayed next. The variables used are:
a [m] Plate lengthb [m] Plate widthh [m] Plate thicknessq [Pa] Distributed loadw [m] Mid-point deflectionm, n  Summation indices

The plate equivalent properties are assumed to be known a priori:
$D_{x}$ [Nm] Plate *x*-direction bending rigidity$D_{y}$ [Nm] Plate *y*-direction bending rigidity$D_{xy}$ [Nm] Plate twist rigidity$\nu_{x}^{b}$, $\nu_{y}^{b}$ Bending Poisson’s ratios$S_{x}$, $S_{y}$ [N/m] Transverse shear rigidity in *x* and *y* directions$E_{x}^{t}$, $E_{y}^{t}$ [Pa] Equivalent plate tension E moduli in *x* and *y* direction$\nu_{x}^{t}$, $\nu_{y}^{t}$ Tension Poisson’s ratios

The *x*, *y* axes origin is set as shown in Figure 2b—at the plate corner.

### 5.2. Small Deflection Coefficient A

The expression in (2) was derived using the Libove and Batdorf NACA Report No. 899 [17].
(2)$$w=\frac{q}{A}=\frac{16q}{\pi^{6}}\sum_{n=1,3,5\ldots }^{\infty }\sum_{m=1,3,5\ldots }^{\infty }\frac{\left[\begin{array}{l} \pi^{4}K_{8}{\left(\frac{m}{a}\right)}^{4}+\pi^{4}K_{9}{\left(\frac{m}{a}\right)}^{2}{\left(\frac{n}{b}\right)}^{2}+\pi^{4}K_{10}{\left(\frac{n}{b}\right)}^{4}\\ -\pi^{2}K_{11}{\left(\frac{m}{a}\right)}^{2}-\pi^{2}K_{12}{\left(\frac{n}{b}\right)}^{2}+K_{13} \end{array}\right]{\left(-1\right)}^{\frac{m+n}{2}-1}}{mn\left[\begin{array}{l} -\pi^{2}K_{1}{\left(\frac{m}{a}\right)}^{6}-\pi^{2}K_{2}{\left(\frac{m}{a}\right)}^{4}{\left(\frac{n}{b}\right)}^{2}-\pi^{2}K_{3}{\left(\frac{m}{a}\right)}^{2}{\left(\frac{n}{b}\right)}^{4}\\ -\pi^{2}K_{4}{\left(\frac{n}{b}\right)}^{6}+K_{5}{\left(\frac{m}{a}\right)}^{4}+K_{6}{\left(\frac{m}{a}\right)}^{2}{\left(\frac{n}{b}\right)}^{2}+K_{7}{\left(\frac{n}{b}\right)}^{4} \end{array}\right]}$$
where
(3)$$K_{1}=\frac{D_{xy}D_{x}}{2S_{y}};\;K_{2}=\frac{D_{xy}D_{x}}{2S_{x}}+\frac{D_{x}D_{y}-D_{xy}D_{x}\nu_{y}^{b}}{S_{y}};\;\;\;K_{3}=\frac{D_{xy}D_{y}}{2S_{y}}+\frac{D_{x}D_{y}-D_{xy}D_{x}\nu_{y}^{b}}{S_{x}};\;K_{4}=\frac{D_{xy}D_{y}}{2S_{x}};\;K_{5}=-D_{x};\;\;K_{6}=-2\left[D_{xy}\left(1-\nu_{x}^{b}\nu_{y}^{b}\right)+D_{x}\nu_{y}^{b}\right];\;K_{7}=-D_{y}\;;\;K_{8}=-\frac{D_{xy}D_{x}}{2S_{x}S_{y}};\;\;\;K_{9}=-\frac{D_{x}D_{y}-D_{xy}D_{x}\nu_{y}^{b}}{S_{x}S_{y}};\;K_{10}=-\frac{D_{xy}D_{y}}{2S_{x}S_{y}}\;K_{11}=\frac{D_{xy}\left(1-\nu_{x}^{b}\nu_{y}^{b}\right)}{2S_{y}}+\frac{D_{x}}{S_{x}};\;K_{12}=\frac{D_{xy}\left(1-\nu_{x}^{b}\nu_{y}^{b}\right)}{2S_{x}}+\frac{D_{y}}{S_{y}};\;\;\;K_{13}=-\left(1-\nu_{x}^{b}\nu_{y}^{b}\right)$$

The complete analysis description is presented in Appendix A.

### 5.3. Large Deflection Coefficient B

As presented above, the coefficient *B* describes the plate’s large deflection response, with the plate being considered as a thin membrane. Normal membranes cannot have a movable (M) BC, so it is complicated to find previous publications displaying an expression for the membrane deflection. Wang and El-Sheikh [6] (p. 816) have analyzed an isotropic rectangular plate and have presented an approximated expression for *q* = *Aw* + *Bw*³ for M BCs: (4)$$q=\frac{\pi^{6}}{64}\left[4Dw{\left(\frac{1}{a^{2}}+\frac{1}{b^{2}}\right)}^{2}+\frac{Ehw^{3}}{4}\left(\frac{1}{a^{4}}+\frac{1}{b^{4}}\right)\right]$$

Modifying this expression to an orthotropic plate and using only the *B* term suggests the expression:(5)$$B=k\cdot h\cdot \left(\frac{{E_{x}}^{t}}{a^{4}}+\frac{{E_{y}}^{t}}{b^{4}}\right)$$
where the coefficient *k* includes all the numerical factors. 

Since (5) is based only on the first term of a multiple term series, the result is not accurate enough to correctly represent the plates response. Therefore, finite element analyses (FEA) of 80 plates with various lengths and widths were performed. The FEA software was Femap with NX-Nastran version 2021.1 with its built-in non-linear static analysis. The FEA results are given in Appendix C, while the FEA information is given in Appendix D. Typical plate arrangements and mid-point load–deflection graphs are shown in Figure 3.

The values of *B* in (1) were calculated with a least-squares regression from the FEA results, and the following expression is suggested for *B*:(6)$$B=h\sqrt{\frac{h}{a+b}}\left(\frac{E_{x}^{t}+E_{y}^{t}}{{\left(a+b\right)}^{4}}\right)K{\left(\frac{a}{b}\right)}^{p}$$

Note that the sum of the moduli and the sum of the length–width represent its averaged values, as the 2 division is included in *K*. The (*a*/*b*) is the aspect ratio of the plate. The *K* and *p* unitless values are *K* = 201.44 and *p* = −0.17165, while the units of the variables are: *B*: [Pa/m^3^], *h*: [m], *a*, *b*: [m], and *E*: [Pa]. Equation (6) describes multiwall plates well, but it is not necessarily suitable for other types of orthotropic plates.

## 6. Vacuum Chamber Test

To check the multiwall plate response to transverse distributed load, a vacuum chamber test was used, as presented in Figure 4:

A 35 mm-thick wooden frame encloses a rectangular space with the required dimensions. The multiwall plate is freely placed on the frame’s edges. A thin plastic sheet covers the entire device and the floor near-by. A variable-speed vacuum cleaner is connected to the internal space through a drilled hole. The vacuum created causes the plastic sheet to seal all air leakages, allowing the vacuum level to gradually increase.

An electronic vacuum sensor measures the vacuum level through another hole in the wooden frame. The sensor is connected to a controller, which displays the data. Additionally, an ultrasonic distance sensor is placed 20 cm above the plate mid-point, measures the plate deflection, and transmits it to the controller. The vacuum units are [Pa], and the distance units are [mm].

The controller has a zero button to zero the vacuum and the distance values as the test starts. During the test, the vacuum level is gradually increased, while both load and deflection are recorded. Various local buckling phenomena are also closely monitored and recorded.

The plate edges are free to move on the wood frame, creating the M movable BC. As the vacuum level increases during the test, the thin plastic sealing sheet experiences tension and presses the plate edge to the wood frame. This may change the BC from SSSS to be somewhat closer to CCCC.

Several tests were performed at the Krumbein Structures Laboratory, Faculty of Aerospace Engineering—Technion. Next, a typical test is presented.

## 7. Results

There were two plates tested: length × width 1.5 × 0.8 m and length × width 0.8 × 1.5 m. The opening dimensions were 0.07 m less, i.e., 1.43 × 0.73 m.

## 8. Plate Details

Type: 10 mm-PC double wallNominal Area Weight 1700 g/m^2^

The measured results were least-squares fitted to the expression *q* = *Aw* + *Bw*³ and were compared to the theoretical curves.

Part of the tested plates data are shown in the following Figure 5, Table 1 and Table 2:

The coefficients *A* and *B* were theoretically calculated with the expressions (2), (6), and they were compared to the measured values. The comparison is shown in Table 2:

As displayed in Table 2, the calculated coefficients comply to the measured one with some differences. 

## 9. FEA Results

To find the response of the plates to transversal uniform load, 80 finite element analyses were performed: 4 plate types with 4 various widths and 5 different lengths (see in Figure 3 one of the tests). The coefficients *A* and *B* for each plate were both theoretically calculated using Equations (2) and (6), respectively, and they were also found from the graphs drawn using Equation (1). A comparison of the theoretically calculated *A* and *B* coefficients and the FEA-measured coefficients is presented in Figure 6. The legend at (b) applies to all other graphs.

The 45° lines represent the location of the perfect agreement between the theory and measurements. As it is shown in Figure 6, a very good agreement between the theory and analysis is found.

## 10. Large Number of Loading Tests

One of the manufacturers of PC multiwall plates is Plazit-Polygal (Plaskolite). During the years 2001–2002, they performed a large number of vacuum loading tests similar to the one described here. Plazit-Polygal has allowed the authors to use the test data for research and publication purposes. This permission is very much appreciated.

From about 250 tests, 120 tests of plates 6–16 mm thick were chosen. Each test has a set of measured load–deflection values for various width–length measurements. The analysis of every test was a linear least-squares regression that calculated the coefficients *A* and *B* in the expression *q* = *Aw* + *Bw*³. The results are listed in Appendix C. The coefficient of determination *R*² (goodness-of-fit) in all tested cases was above 0.99. These very good fits support the validity of the suggested *A, B* large deflection approximation (Equation (1)).

The equivalent elastic constants and moduli of the plates were measured and given with the test data in Appendix B. This information allows for the calculation of the *A, B* coefficients according to the theory above. The measured and calculated *A, B* coefficients are compared in Figure 7 and Figure 8, where the unit of *A* is [Pa/m] and the unit of *B* is [Pa/m^3^].

In Figure 7 and Figure 8, the horizontal axes are the *A* and *B* measured coefficients, while the vertical axes are the theoretically calculated coefficients. The 45° lines represent the location of the perfect agreement between the theory and measurements. The various colors are for various plate widths [m], as shown in the Figure 6b graph legend.

Generally, the theory–measurement agreements are better for the coefficient *A* over the coefficient *B*, and they are also better for higher thickness over lower thickness. 

Since Polygal’s tests were performed more than 20 years ago, the tested samples are not available for verification anymore. Many dimensions of the plates were missing in the records, so the standard values were taken from the plate’s data sheets. Nevertheless, the actual real plate dimension values almost always deviate from the standard values, as may occur in real manufacturing. These deviations can be rather significant and, for sure, influence on the plate rigidities. This is probably the main source for the inconsistencies in the reported data.

As the vacuum was measured at that time with a water manometer, which is not accurate enough, it is very possible that errors do exist in the data. These errors can be seen in Appendix C, Table A4, Table A5, Table A6 and Table A7, where the measured coefficients *A* should monotonically increase while the length values decrease, but in several cases, they unexpectedly decrease.

By being very comprehensive, with a large number of length/width combinations, the understanding of the plate response to transverse pressure still has some value, besides proving the *q* = *Aw* + *Bw*³ response.

Note that the last plate test (10 mm Plate Faculty Lab) was performed more recently, under a better-controlled environment, and therefore, it presents a more accurate agreement.

## 11. Conclusions

Large deflection of multiwall plates, under distributed transverse pressure and SSSS-M boundary conditions, has been found to comply with the *q* = *Aw* + *Bw*³ approximation rule with very good *R*² (goodness-of-fit) values.

The suggested expressions for *A* and *B* coefficients have good agreement with FEA results, while in actual tests, they appear to be more applicable for thicker plates.

The deviations in the presented experimental loading data may relate to the measurement technique. Good laboratory practice would lead to more accurate results.

## Figures and Tables

**Figure 1 materials-16-02041-f001:**
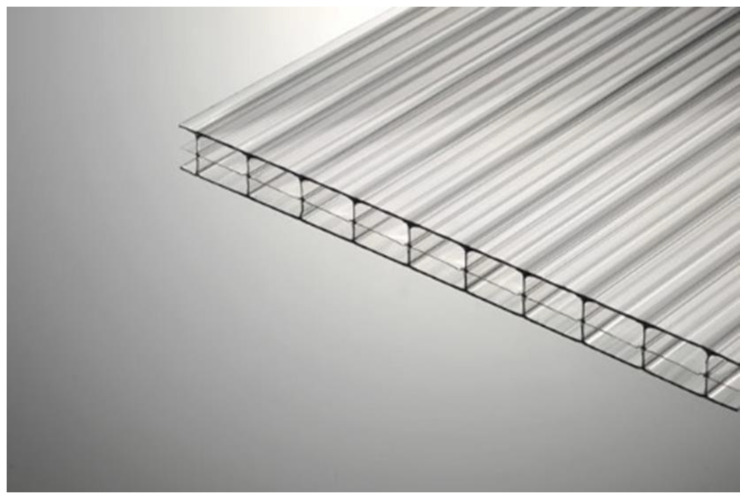
A 16 mm Multiwall PC Plate.

**Figure 2 materials-16-02041-f002:**
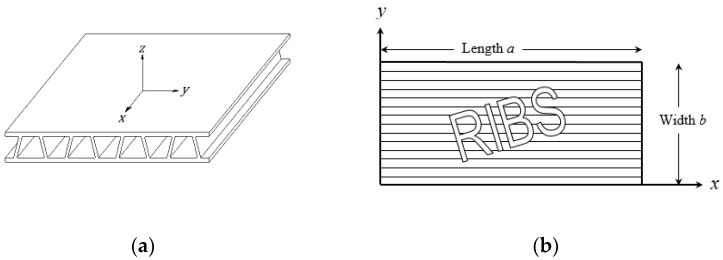
Axes directions: (**a**) 3D view trapezoid plate, (**b**) 2D view ribbed plate.

**Figure 3 materials-16-02041-f003:**
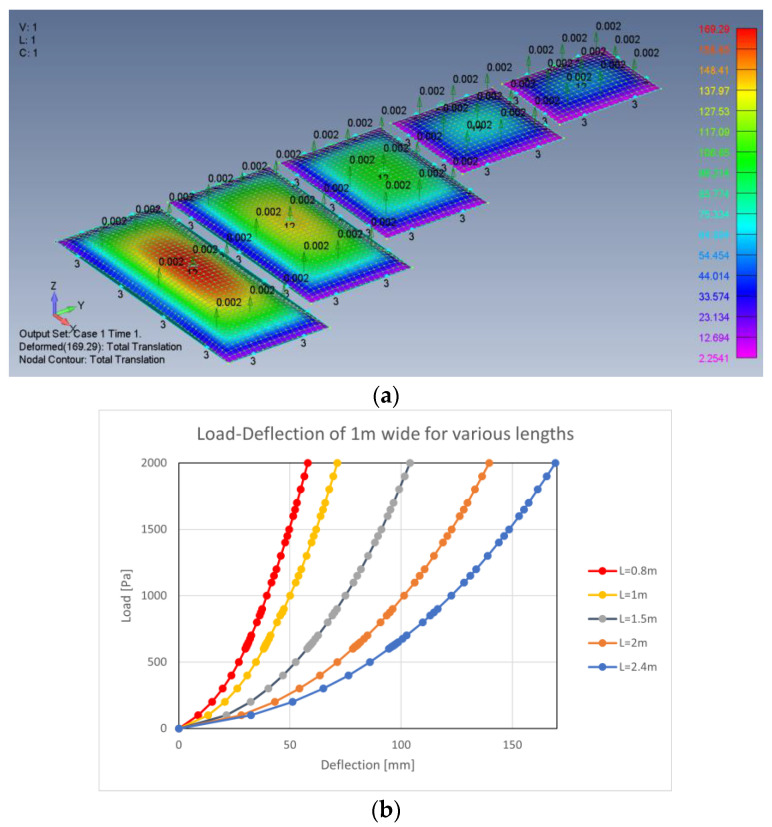
FEA deformation map (**a**) and load–deflection curves (**b**).

**Figure 4 materials-16-02041-f004:**
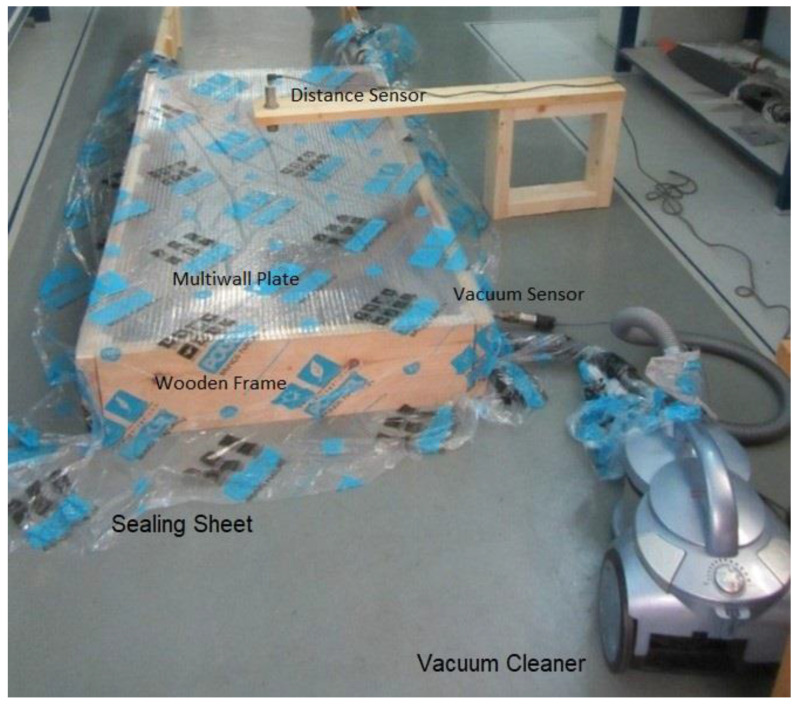
The vacuum chamber test set.

**Figure 5 materials-16-02041-f005:**
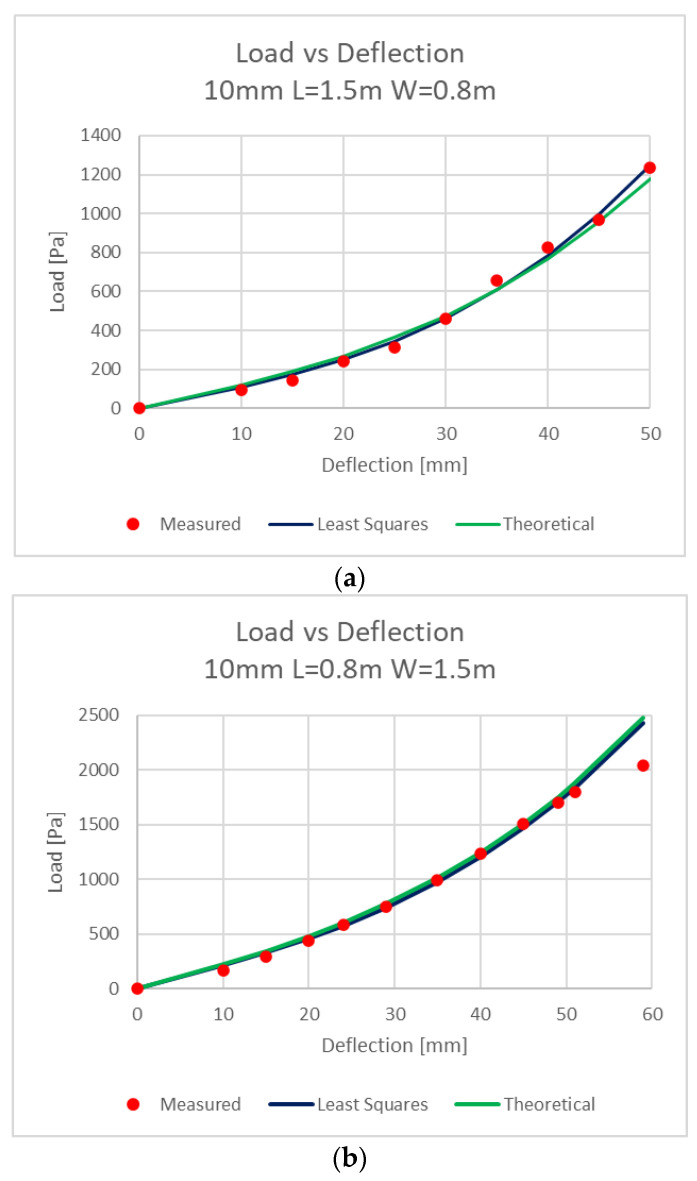
Load–Deflection graphs: (**a**) length 1.5 m, width 0.8 m; (**b**) length 0.8 m, width 1.5 m.

**Figure 6 materials-16-02041-f006:**
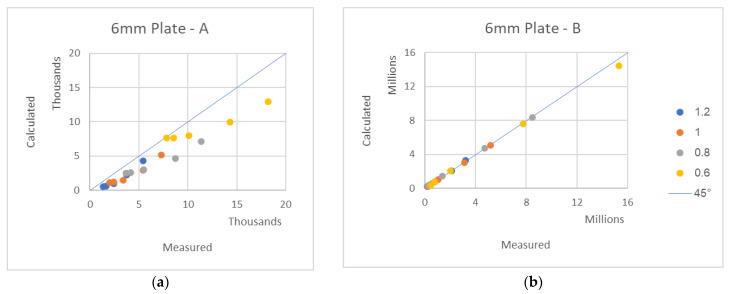
FEA-measured-calculated *A*, *B* expressions—a comparison of various plates: (**a**) 6 mm A, (**b**) 6 mm B with legend, (**c**) 8 mm A, (**d**) 8 mm B, (**e**) 10 mm A, (**f**) 10 mm B, (**g**) 16 mm A, (**h**) 16 mm B.

**Figure 7 materials-16-02041-f007:**
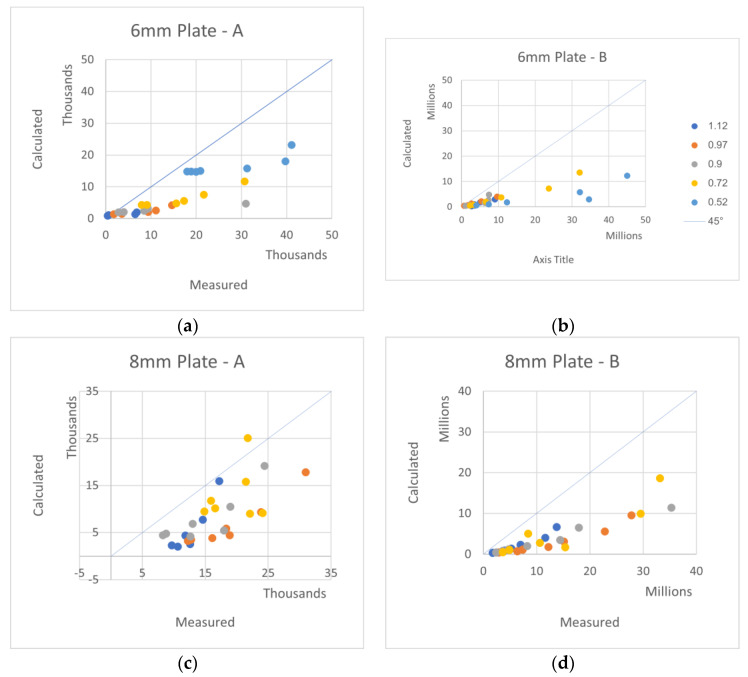
Measured–Calculated *A*, *B* constants—a comparison of various plates. (**a**) 6 mm A, (**b**) 6 mm B with legend, (**c**) 8 mm A, (**d**) 8 mm B, (**e**) 10 mm A, (**f**) 10 mm B, (**g**) 16 mm A, (**h**) 16 mm B.

**Figure 8 materials-16-02041-f008:**
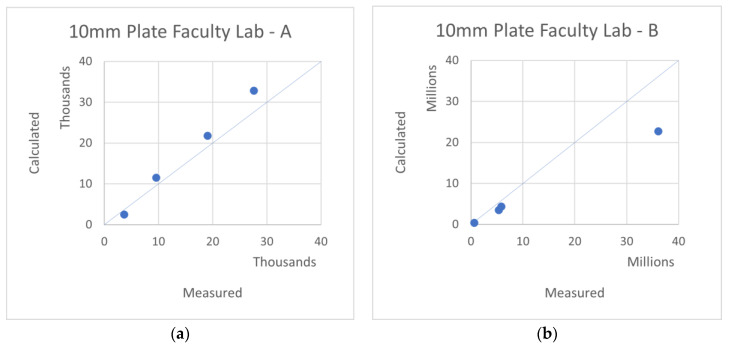
Measured–Calculated *A*, *B*—a comparison of various plates performed recently at the Aerospace Lab. (**a**) 10 mm A, (**b**) 10 mm B.

**Table 1 materials-16-02041-t001:** The elastic constants of this multiwall plate.

Thickness	*h*	[mm]	10
Area Weight	*W*	[g/m²]	1713
Walls Thickness	*t_w_*	[mm]	1.154
Equivalent *G*	*G_eq_*	[MPa]	100.348
For small deflection coefficient *A*:			
	*D_x_*	[Nm]	70.12106
Bending:	*D_y_*	[Nm]	54.10356
	*D_xy_*	[Nm]	8.362
Shear:	*S_x_*	[N/m]	59,890.03
	*S_y_*	[N/m]	1662.076
	*ν_x_^b^*		0.38
	*ν_y_^b^* = *D_y_*/*D_x_* ∗ *ν_x_^b^*		0.293
For large deflection coefficient *B*:			
	*E_x_^t^*	[MPa]	342.60
Tension:	*E_y_^t^*	[MPa]	276.96
	*ν_x_^t^*		0.38
	*ν_y_^t^* = *E_y_^t^*/*E_x_^t^* ∗ *ν_x_^t^*		0.307

**Table 2 materials-16-02041-t002:** Coefficients results.

Coefficient	Length 1.5 m, Width 0.8 m			Length 0.8 m, Width 1.5 m		
	Measured	Theoretical	% Difference	Measured	Theoretical	% Difference
*A* [Pa/m]	9602	11,487 (2)	16.4%	19,061	21,805	12.6%
*B* [Pa/m³]	5,370,163	3,475,816 (6)	35.3%	5,858,384	4,378,294	25.3%

## Data Availability

The data presented in this study are available in this article.

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
