# Peer review of "Multiwall Rectangular Plates under Transverse Pressure—A Non-Linear Experimental and Numerical Study"

_materials, 2023, doi:10.3390/ma16052041_

Round 1
Reviewer 1 Report
In the present manuscript, large deflection analysis of rectangular plates under transverse pressure have been discussed by Föppl von Kármán equations and analytical method. Also, resulta validate the non-linear relationship between the pressure and the lateral displacement of the multiwall plate, a vacuum chamber loading test is used to measure the plate's response, with a large number of plates and length width combinations.
Manuscript can be accepted after minor revision:
1 Authors are encouraged to show the main practical applications of the solved problem in detail. The authors should point out the deficiencies of the published studies to make a bridge to introducing the novelty of their work.
The current references are appropriate. However, the reference list is too short. Consequently, some new references about the research
Geometrically nonlinear dynamic and static analysis of shallow spherical shell resting on two-parameters elastic foundations. International Journal of Pressure Vessels and Piping 113, 1-9 (2014)
Beatty MF. Introduction to Nonlinear Elasticity. Nonlinear Eff Fluids Solids 1996: 13–112. https://doi.org/10.1007/978-1-4613-0329-9_2.
Ogden RW. Non-linear elastic deformations 1997:532.
Author Response
Reviewer #1: Comments and Suggestions for Authors
The authors would like to thank the reviewer for his constructive comments.
Our reply to reviewer's comments, are highlighted in yellow.
In the present manuscript, large deflection analysis of rectangular plates under transverse pressure have been discussed by Föppl von Kármán equations and analytical method. Also, resulta validate the non-linear relationship between the pressure and the lateral displacement of the multiwall plate, a vacuum chamber loading test is used to measure the plate's response, with a large number of plates and length width combinations.
Manuscript can be accepted after minor revision:
- Authors are encouraged to show the main practical applications of the solved problem in detail. The authors should point out the deficiencies of the published studies to make a bridge to introducing the novelty of their work.
An application paragraph has been added in the revised manuscript with the relevant novelty of the present study.
- The current references are appropriate. However, the reference list is too short. Consequently, some new references about the research
Geometrically nonlinear dynamic and static analysis of shallow spherical shell resting on two-parameters elastic foundations. International Journal of Pressure Vessels and Piping 113, 1-9 (2014)
Beatty MF. Introduction to Nonlinear Elasticity. Nonlinear Eff Fluids Solids 1996: 13–112. https://doi.org/10.1007/978-1-4613-0329-9_2.
Ogden RW. Non-linear elastic deformations 1997:532.
The above references have been added with other more references being included in the manuscript.

Reviewer 2 Report
The paper is quite short and poor in the description of methodology and conclusions of the work. The analytical expression proposed seems too simplistic with respect to the complexity of the problem considered and it risks to fail in most of the study cases. Indeed, the results show a discrepancy with experimental results, although the authors justify it with the low quality of practices in laboratory (they should prove that). Even the comparison with numerical results is not very rielable since the details of the FEA models are not provided. In addition to the comments above, the authors use effective properties of the plate without explaining their derivation.
Author Response
Reviewer # 2 : Comments and Suggestions for the Authors
The authors would like to thank the reviewer for his constructive comments.
Our reply to reviewer's comments, are highlighted in yellow.
The paper is quite short and poor in the description of methodology and conclusions of the work. The analytical expression proposed seems too simplistic with respect to the complexity of the problem considered and it risks to fail in most of the study cases.
The separation into the small deflection and membrane model was suggested long time ago, and although being relatively simple, it has a potential to represent the plate nonlinear behavior quite good. Finding the constants of the nonlinear curve, lateral displacement vs. applied pressure using the plate's elastic constants is quite challenging as can be seen in the paper. The authors investigated both the experimental and the numerical results and provided , to our belief, for the first time such a comparison.
Indeed, the results show a discrepancy with experimental results, although the authors justify it with the low quality of practices in laboratory (they should prove that).
A proof explanation has been added in the revised manuscript.
Even the comparison with numerical results is not very rielable since the details of the FEA models are not provided.
The details of the FEA were added in the revised version of the manuscript.
In addition to the comments above, the authors use effective properties of the plate without explaining their derivation.
The effective properties are assumed to be known a priori, where the methods to get it are described in detail in reference [16] of the manuscript.

Reviewer 3 Report
The authors should discuss the obtained results and explain the outcomes in the obtained tables.
The authors should give a suitable background of the discussed methodology and explain the novelty of the new study.
Comparisons to other approaches should be made.
Author Response
Reviewer # 3 :Comments and Suggestions for Authors
The authors would like to thank the reviewer for his constructive comments.
Our reply to reviewer's comments, are highlighted in yellow.
The authors should discuss the obtained results and explain the outcomes in the obtained tables.
Explanations of the results in the various tables have been added, including description of deviations of the data quality.
The authors should give a suitable background of the discussed methodology and explain the novelty of the new study.
A novelty-dedicated paragraph has been added.
Comparisons to other approaches should be made.
To the best knowledge of the authors, there are no other approaches in the literature for presenting the curve load vs. lateral displacement for a lateral loaded plate in its nonlinear regime, beside those presented in the literature.

Reviewer 4 Report
Dear Authors,
My remarks:
1. Introduction, literature review, motivation, and explanation of novelty is insufficiently discussed.
2. The structure of the paper should be revised with respect to the recommendation given in the template.
3. Figures are blurred.
4. The numbering of equation should be checked (missing eq. 3).
5. The nomenclature should be rather given in separate section.
6. The equation 2 seems to be shortened and adapted solution given in the reference published in 1948 [1]. Please provide information what is new in a proposed mathematical model.
7. Did the solutions for parameter B (eqs 4-6) are found by Authors or they are taken from literature? Please provide references.
8. Line 139 “In this FEA, all membrane buckling (wrinkling) effects were disabled such…” In such situation references to such analysis or more detailed description of FEM model should be provided (solver, mesh, elements, boundary conditions, etc.).
9. Figures 6 and 7. The figures are really difficult to read due to missing description in legend. Moreover, it seems that in all cases the calculated parameter B is strongly underestimated in all cases. Similar situation is with parameter A. The worst results are obtained for plates with thickness below 10 mm. Basing on such verification it is difficult to evaluate the usefulness and correctness of the proposed model. An explanation that “The deviations in the presented experimental loading data may relate to the measurement technic. Good laboratory practice would lead to more accurate results.” (phrase taken from conclusions) is insufficient and rather indicates the incorrect approach of the authors to the verification of the model. The Authors should make verification of the model with the use of the proper experimental data. The application of 20-years old data which was measured without accurate equipment and which cannot be validated is completely unjustified.
10. Appendixes : missing units. i.e. units of Length and width.
[1] Libove C.; Batdorf S.B., A General Small-Deflection Theory for Flat Sandwich Plates, NACA Report No. 899, 1948; pp.139-156
Kind Regards,
Author Response
Reviewer # 4 :Comments and Suggestions for Authors
The authors would like to thank the reviewer for his constructive comments.
Our reply to reviewer's comments, are highlighted in yellow.
Dear Authors,
My remarks:
- Introduction, literature review, motivation, and explanation of novelty is insufficiently discussed.
The literature list has been expanded. A motivation paragraph has been added. A novelty-dedicated paragraph has been added – the 2nd paragraph after Figure 1.
- The structure of the paper should be revised with respect to the recommendation given in the template.
The structure of the paper is now according to the template.
- Figures are blurred.
The quality was improved and we hope it should be good by now.
- The numbering of equation should be checked (missing eq. 3).
The topic was corrected.
- The nomenclature should be rather given in separate section.
Various acronyms are explained in its first appearance in the paper.
- The equation 2 seems to be shortened and adapted solution given in the reference published in 1948 [1]. Please provide information what is new in a proposed mathematical model.
Equations (2) and (3) are the solution of the partial differential equation (PDE) given in [1]. The in-plane terms were removed from the PDE since they are zero in our case. The solution in (2,3) includes the BCs and the distributed load. The entire solution is given in Appendix A. Although the solution method is a basic technique in linear PDEs, this specific solution could not be found in the literature and therefore been presented in the paper.
- Did the solutions for parameter B (eqs 4-6) are found by Authors or they are taken from literature? Please provide references.
Expression (4) was presented by Wang and El-Sheikh [6] for isotropic material. It was modified to an orthotropic material by the authors. Yet, a mistake has been occurred during this modification. A corrected expression exists now in (5). As a result, coefficient k value has also been changed from the wrong value 0.38911 to a correct value 0.457 .
- Line 139 “In this FEA, all membrane buckling (wrinkling) effects were disabled such…” In such situation references to such analysis or more detailed description of FEM model should be provided (solver, mesh, elements, boundary conditions, etc.).
Detailed information and a figure have been added to the FEA section.
- Figures 6 and 7. The figures are really difficult to read due to missing description in legend.
Legend is provided in Figure 6b and removed from the other for clarity.
Moreover, it seems that in all cases the calculated parameter B is strongly underestimated in all cases. Similar situation is with parameter A. The worst results are obtained for plates with thickness below 10 mm. Basing on such verification it is difficult to evaluate the usefulness and correctness of the proposed model. An explanation that “The deviations in the presented experimental loading data may relate to the measurement technic. Good laboratory practice would lead to more accurate results.” (phrase taken from conclusions) is insufficient and rather indicates the incorrect approach of the authors to the verification of the model. The Authors should make verification of the model with the use of the proper experimental data. The application of 20-years old data which was measured without accurate equipment and which cannot be validated is completely unjustified.
It is obvious that there is a deviation between the numerical model and the results. Yet, the first assumption that the plate has a polynomial response q=Aw+Bw³ is fully supported in the results. The last test done in the faculty lab has a much better agreement, and charts have been added. The gaps that still exist may encourage other researchers to perform more accurate tests on these type plates in parallel to improving the numerical model.
- Appendixes : missing units. i.e. units of Length and width.
Corrected.
[1] Libove C.; Batdorf S.B., A General Small-Deflection Theory for Flat Sandwich Plates, NACA Report No. 899, 1948; pp.139-156

Round 2
Reviewer 2 Report
The paper can be accepted.
Reviewer 3 Report
The authors have modified the manuscript and in my opinion it is suitable for publishing.
Author Response
The authors would like to thank the reviewer for his constructive comments.
Our reply to reviewer's comments, are highlighted in yellow.
The authors have modified the manuscript and in my opinion, it is suitable for publishing.
The authors are grateful to the reviewer decision.

Reviewer 4 Report
Dear Author,
1. In my previous revision I indicated that there are significant differences between calculated and measured values of parameters A and B. Both parameters are the most important parts of the proposed mathematical formulation. Based on the data given in Figures 6 and 7 it can be seen that:
- the calculated parameter B is highly underestimated with respect to the measured one in the whole range for plates with thicknesses 6, 8, 10 (figure 6f), and 16 mm.
- in my opinion, the “faculty lab tests” also suggest that the calculated parameter B is highly underestimated. The good agreement is observed only for 3 points with small values, but for higher values again there is a visible high underestimation. If one compares figure 6f and 7j then it can be seen that in both cases there is an important problem with the convergence of parameter B for values higher than 0.01.
-the calculated parameter A is highly underestimated for plates with thicknesses 6, and 8 mm.
-good convergence of parameter A was obtained for plates with thicknesses 10 and 16 mm.
The results obtained for parameter A also emphasize that the formulation for the calculation of parameter B is probably wrong.
Based on the above conclusion as well as on the Authors response, still, I was not convinced about the correctness of the presented method. In such a situation further studies are required.
- more points for parameter B within a range 0.01-0.04 should be tested in the laboratory,
- FEM verification of the examples in which there is no agreement between calculated and measured parameters A and B in order to find the reason for such differences,
-modification of the mathematical model (influence of thickness, and non-known reason in the case of parameter B).
2. Introduction is still not sufficient. The new part (lines 76-91) should be rather given in the introduction.
3. The grammar/style of changed or introduced new parts is poor. There are also mistakes such as ”1 mm thick” (line 172)
4. figure captions are still not correct, i.e. Fig 3b: it should be given that it is a deformation map; Figure 5 should be divided into Figures 5a and 5b, etc.
5. Lines 231-235: It is not clear. It should be specified what plates were tested i.e. 1.5x0.8. Were the plates with dimensions 1.5x1.5 and 0.8x0.8 tested?
6. Figure 2: What about FEM analysis? Mainly in the case in which higher differences occurred.
7. References in my opinion are not formatted correctly.
8. What is the reason for changing the calculated values of parameters A and B in appendix B? All parameters B have been changed with respect to the previous version and the large number of parameter A was changed.
Kind Regards,
Author Response
The authors would like to thank the reviewer for his constructive comments.
Our reply to reviewer's comments, are highlighted in yellow.
- In my previous revision I indicated that there are significant differences between calculated and measured values of parameters A and B. Both parameters are the most important parts of the proposed mathematical formulation. Based on the data given in Figures 6 and 7 it can be seen that:
- the calculated parameter B is highly underestimated with respect to the measured one in the whole range for plates with thicknesses 6, 8, 10 (figure 6f), and 16 mm.
- in my opinion, the “faculty lab tests” also suggest that the calculated parameter B is highly underestimated. The good agreement is observed only for 3 points with small values, but for higher values again there is a visible high underestimation. If one compares figure 6f and 7j then it can be seen that in both cases there is an important problem with the convergence of parameter B for values higher than 0.01.
-the calculated parameter A is highly underestimated for plates with thicknesses 6, and 8 mm.
-good convergence of parameter A was obtained for plates with thicknesses 10 and 16 mm.
The results obtained for parameter A also emphasize that the formulation for the calculation of parameter B is probably wrong.
Based on the above conclusion as well as on the Authors response, still, I was not convinced about the correctness of the presented method. In such a situation further studies are required.
- more points for parameter B within a range 0.01-0.04 should be tested in the laboratory,
- FEM verification of the examples in which there is no agreement between calculated and measured parameters A and B in order to find the reason for such differences,
-modification of the mathematical model (influence of thickness, and non-known reason in the case of parameter B).
The authors completely agree with the essence of the above comment. Yet, it is important to notice that the expression of B in the article is taken from Wang et al [6, which is based only on the first term of their analysis. Using more terms might improve the accuracy. The reason it was not done in the present study is that it requires rather complicated iterative numerical computation. This is the first attempt to verify this theoretical expression of B with experimental results. Obviously, the results are not very good, so further future improvement of the expression is required. Please note that the numerical coefficient k has a linear influence on B. Currently, the value of k is taken from the membrane FEA as described in the article. It was possible to calculate the value of k to best fit the experimental data and improve the results. In this situation however, we could have a disagreement with the FEA calculation. Therefore, future work should rely on more accurate test results and possibly improve the theoretical analysis.
We have checked other researchers' reports on parameters A and B (see below). The inclusion of this literature survey was beyond the scope of the present article. Yet, just to demonstrate the large scattering of the experimental results from different sources the below table is presented:
For a square isotropic thin plate:
|
No. |
Ref. |
Source |
||
|
0 |
- |
Navier's linear CPT |
246.16 |
— |
|
1 |
1 |
Yankelevsky D. et al |
389.64 |
6.3238 |
|
2 |
2 |
Walter D. Pilkey |
240.35 |
7.4723 |
|
3 |
3 |
Levy Samuel |
240.475 |
8.9562 |
|
4 |
4 |
Ishizaki Hatsuo |
220 |
1.7787 |
|
5 |
5 |
ASTM E 1300 |
243.2 |
2.29 |
|
6 |
6 |
Scholes A. |
260.3 |
3.547 |
|
7 |
7 |
Kaiser Rudolf |
251.98 |
3.366 |
|
8 |
8 |
Chia Chuen-Yuan |
240.35 |
3.7972 |
|
9 |
9 |
Brown J. C. |
307.9 |
2.799 |
References:
[1] Yankelevsky D., Feldgun V., Karinsky Y., The mechanical behavior of glass plates,
National Building Research Institute - Technion (Hebrew Language) (2017)
[2] Walter D. Pilkey, Formulas for Stress, Strain, and Structural Matrices,
John Wiley & Sons, Inc. (2005)
[3] Levy S., Bending of Rectangular Plates with Large Deflections, NACA Report 737, (1941)
[4] Ishizaki Hatsuo, On the Large Deflections of Rectangular Glass Panes under Uniform Pressure, Bulletin of the Disaster Prevention Research Institute, 22(1): 1-7, Kurenai Kyoto University (1972)
[5] ASTM E 1300, Standard Practice for Determining Load Resistance of Glass in Buildings (2009)
[6] Scholes A., Bernstein E. L., Bending of Normally Loaded Simply Supported Rectangular Plates in the Large-Deflection Range,
The Journal of Strain Analysis for Engineering Design 1969 4: 190, (1969)
[7] Kaiser Rudolf, Rechnerische und experimentelle Ermittlung der Durchbiegungenund Spannungen von quadratischen Platten bei freier Auflagerung an den Rändern, gleichmäßig verteilter Last und großen Ausbiegungen,
ZAMM: Zeitschrift für angewandte Mathematik und Mechanik, Band 16, Heft 2, April (1936)
[8] Chia Chuen-Yuan, Nonlinear Analysis of Plates, McGraw-Hill, Inc. (1980)
[9] Brown J. C., Harvey J. M., Large Deflections of Rectangular Plates Subjected to Uniform Lateral Pressure and Compressive Edge Loading,
Journal Mechanical Engineering Science, Vol 11 No 3 (1969)
- Introduction is still not sufficient. The new part (lines 76-91) should be rather given in the introduction.
The Introduction section has already several sub-sections, which are: The first part of the introduction (general presentation of the topic), Multiwall Plates, Axes system and Boundary conditions (BCs).
Lines 76-91 belong to the Multiwall sub-section. Moving it to the first section will cut its context.
- The grammar/style of changed or introduced new parts is poor. There are also mistakes such as ”1 mm thick” (line 172)
The text has been corrected.
- figure captions are still not correct, i.e. Fig 3b: it should be given that it is a deformation map; Figure 5 should be divided into Figures 5a and 5b, etc.
The captions issue has been corrected in the text.
- Lines 231-235: It is not clear. It should be specified what plates were tested i.e. 1.5x0.8.
Done in the text.
Were the plates with dimensions 1.5x1.5 and 0.8x0.8 tested?
1.5x1.5 plate was not tested. 0.8x0.8 and 1.2x2.4 plates were also tested but discarded due to too big difference between the two load directions, probably caused by a wrong sensor calibration.
- Figure 2: What about FEM analysis? Mainly in the case in which higher differences occurred.
It is difficult to understand this point, Figure 2 shows the axes system, not FE. Assuming the reviewer meant FEA in Figure 3, it is still not clear.
- References in my opinion are not formatted correctly.
The references list has been reformatted according to Materials requirements.
- What is the reason for changing the calculated values of parameters A and B in appendix B? All parameters B have been changed with respect to the previous version and the large number of parameter A was changed.
The B values were changed after the correction of the k value. All B values were re-calculated.
